# A Data Processing and Distribution System Based on Apache NiFi

**Karol Wnęk and Piotr Boryło \***

AGH University of Science and Technology, Institute of Telecomunications, 30-059 Kraków, Poland
\* Correspondence: piotr.borylo@agh.edu.pl

**Abstract:** The monitoring of physical and logical networks is essential for the high availability of 5G/6G networks. This could become a challenge in 5G/6G deployments due to the heterogeneity of the optical layer. It uses equipment from multiple vendors, and, as a result, the protocols and methods for gathering monitoring data usually differ. Simultaneously, to effectively support 5G/6G networks, the optical infrastructure should also be dense and ensure high throughput. Thus, vast numbers of photonic transceivers operating at up to 400 Gbps are needed to interconnect network components. In demanding optical solutions for 5G and beyond, enterprise-class equipment will be used—for example, high-capacity and high-density optical switches based on the SONiC distribution. These emerging devices produce vast amounts of data on the operational parameters of each optical transceiver, which should be effectively collected, processed, and analyzed. The aforementioned circumstances may lead to the necessity of using multiple independent monitoring systems dedicated to specific optical hardware. Apache NiFi can be used to address these potential issues. Its high configurability enables the aggregation of unstandardized log files collected from heterogenous devices. Furthermore, it is possible to configure Apache NiFi to absorb huge data streams about each of the thousands of transceivers comprising high-density optical switches. In this way, data can be preprocessed by using Apache NiFi and later uploaded to a dedicated system. In this paper, we focus on presenting the tool, its capabilities, and how it scales horizontally. The proven scalability is essential for making it usable in optical networks that support 5G/6G networks. Finally, we propose a unique optimization process that can greatly improve the performance and make Apache NiFi suitable for high-throughput and high-density photonic devices and optical networks. We also present some insider information on real-life implementations of Apache NiFi in commercial 5G networks that fully rely on optical networks.

**Keywords:** Apache NiFi; big data; data processing; ETL

## 1. Introduction

The optical layer is a critical part of the 5G/6G infrastructure. As interconnected optical devices ensure connectivity for all of the upper layers, any issues will impact every aspect of the mobile network environment, e.g., the operation of a core network and data networks handling user traffic. Thus, it is critical to ensure reliable and effective monitoring of optical networks. However, it is simultaneously challenging because equipment from multiple vendors are used, and, as a result, the protocols and methods for gathering monitoring data usually differ. Furthermore, the optical infrastructure in 5G/6G networks is dense—for example, it comprises a large number of photonic transceivers operating on up to 400 Gbps interfaces. These enterprise-class devices produce vast amounts of data on the operational parameters of each optical transceiver, which should be effectively collected, processed, and analyzed. For example, in a commercial 5G network, the daily volume of data related to the state of the network may be around 864 GB on average (internal statistics of the company in which one of the co-authors works).

Moreover, the ever-increasing computerization and the development and popularization of services provided via the Internet result in the generation and consumption of more and more digital data. Mobile networks contribute to these statistics to a greater extent every year [1]. Statista examined the market as a whole and the volume of information created, modified, and consumed on an annual basis. The report stated that in 2010, there were 2 zettabytes of information globally; in 2015, there were already 15.5 ZB, and in 2020, the annual value increased by another 50 ZB [2]. According to forecasts, the 180 ZB limit will be exceeded in 2025. This growth rate, combined with the volume of data, is motivating the emergence of new concepts. For example, Big Data is the analysis of datasets while looking for correlations and formulating conclusions based on these analyses. The monitoring systems for 5G/6G networks are expected to cover more and more data to ensure comprehensive analysis of all events and issues. The estimated value of the industry related to data in 2023 could be as high as USD 103 billion [3].

Manual analysis of datasets that are not even larger than a few gigabytes can be too time-consuming. Therefore, proper tools are needed to efficiently manipulate data. One of the available solutions is NiFi, which was developed by the Apache Foundation. It automates data processing and the distribution of data. Instead of using multiple independent monitoring systems dedicated to specific optical hardware in a 5G/6G infrastructure, Apache NiFi can be deployed to absorb huge data streams originating from thousands of transceivers comprising high-density optical switches. Additional streams may also originate from any upper-layer devices, services, and infrastructures. Apache NiFi can be configured to aggregate unstandardized log files collected from heterogeneous devices. In this way, the data can be preprocessed by using Apache NiFi and later uploaded to a dedicated system. Efficient data processing can be further used to trigger, for example, resiliency mechanisms [4] or advanced prediction models after node failures [5].

In this work, we focus on the following open issues:

- monitoring a heterogeneous multi-vendor optical network by using a single system;
- efficient absorption of huge data streams about each of the thousands of transceivers comprising high-density optical switches;
- robust preprocessing of the collected datasets for further analysis;
- time-efficiently reflecting changes in the monitored infrastructure.

The main contributions of our work that address these issues are as follows:

- We present the Apache NiFi tool with a special focus on its capabilities of collecting data from heterogeneous optical devices;
- We propose a unique optimization process that can greatly improve the performance and make Apache NiFi suitable for high-throughput and high-density photonic devices and optical networks;
- We conduct experiments that prove the efficiency and scalability of the proposed system;
- We present some insider information on real-life implementations of Apache NiFi in commercial 5G networks that fully rely on optical networks.

The rest of this paper is organized as follows. The next section presents the fundamentals of the Apache NiFi tool. Section 3 refers to the current state of the art with a special focus on monitoring optical networks in a 5G/6G infrastructure by using Apache NiFi. Section 4 describes a commercial deployment of NiFi aimed at monitoring a 5G optical network. Section 5 focuses on our original research—namely, the research environment, the proposed optimization of Apache NiFi for 5G optical networks, and the collected results, along with a comprehensive analysis. Section 6 concludes the paper.

## 2. Background

Apache NiFi can be defined as an Extract, Transform, and Load (ETL) software. A typical usage scenario is that of taking raw data and then adapting them to a format supported by the recipient. An example would be querying a physical switch deployed

in a 5G/6G network about its optical network transceivers and uploading the results to a database for further analysis. Individual steps are implemented in a flow-based manner. This means that the configuration takes the form of a processing path consisting of specialized modules that are responsible for changing the data. The modules are connected to each other by queues that allow the storage of quanta of data. The entire system operates asynchronously, and the processes in the modules can be executed in parallel, as they are independent of each other. More than 250 pre-defined FlowFile Processors are available to users, and it is also possible to define new modules for non-standard tasks. A major advantage of Apache NiFi is the ability to combine several computing nodes into a single cluster. This increases the performance without the need for changes to the processing path. This functionality may become useful when deploying a new site in a 5G/6G network that may be composed of numerous optical devices reporting their operational parameters. The basic terminology and parameters related to Apache NiFi are outlined below.

A **FlowFile** represents a single quantum of data consisting of two elements: content and attributes (metadata in the form of key–value pairs).

A **FlowFile Processor** is a module that performs a specific task on the input data and sends the result to one or more outputs. It has full access to the attributes and content of the FlowFile. At any given time, it can handle individual FlowFiles or groups of FlowFiles. Examples of predefined modules are *LogMessage* for writing messages to the event log, *UpdateAttribute* for modifying attributes, and *ReplaceText* for changing the content of a FlowFile. For example, these modules can be used to parse and handle SNMP (Simple Network Management Protocol) events generated by optical devices deployed in a 5G/6G network.

A **Connection** is a link between Processors that also acts as a queue. It enables data exchange between blocks operating at different speeds. In addition, it supports FlowFile prioritization and the distribution of FlowFiles across cluster nodes for load-balancing purposes.

**Concurrent Tasks** are the numbers of threads assigned to given FlowFile Processors within a single node—by default, this is equal to 1.

**Run Duration** determines how long the FlowFile Processors can serve another FlowFile without saving changes. Larger values provide higher throughput, but at the cost of increased latency. The default value is 0 ms, which means that the Processor separately saves the changes made for each FlowFile.

The use case presented in our work originates from the deployment of NiFi for a commercial 5G network (one of the co-authors works in a company that develops and maintains NiFi for one of the leading operators of a 5G network in the Far East region). To ensure the confidentiality of commercial solutions, some data have been censored. Furthermore, the quality of the provided images might be unsatisfactory because the production environment is maintained by the client themselves, and there is limited access to it due to security restrictions.

One of the requirements for a 5G network operator is the ability to build a graphical representation of the physical infrastructure. To provide a valid solution, the whole network was properly modeled. Starting from the location of the hardware, we ended with the optical fibers and transceivers interconnecting resources. Apache NiFi was used as middleware between various sensors/devices and the relational database that stored all necessary data. NiFi also provided the data to the rest of the system. The details are presented in Section 4.

## 3. State of the Art

The problem of log processing in a 5G network is present in the literature. In a very recent work [6], the authors highlighted the challenges of building a security operation center (SOC) for 5G technology. They emphasized the same issues as those mentioned in our work: high network capacity, low latency, and a large number of heterogeneous devices. To solve these problems, the authors proposed a solution for collecting, storing,

and visualizing logs from a 5G access network and end devices. The solution was based on the Message Queuing Telemetry Transport (MQTT) protocol for collecting the data. However, contrary to Apache NiFi, MQTT is not able to preprocess data. For this purpose, it should be integrated with the Elastic Stack. Monitoring of a 5G access network for the purpose of building a comprehensive SOC solution was also the objective of a thesis [7]. The author focused especially on challenges related to network slicing. The main advantage of this work was its very detailed explanation of how the monitoring part was implemented by using the public–subscribe paradigm. The proposed solution was also integrated with the Elastic Stack for further processing.

Monitoring and collecting logs in the 5G infrastructure is also part of the QoS assurance process. In [8], the authors employed artificial intelligence to ensure proper quality of service in 5G. Again, the MQTT protocol was used to transport the data. For the purpose of log collection, they assumed the use of generic ETL software, though Apache NiFi is a flagship example of this class of tool. An interesting 5G/6G testbed was built and described in [9]. Monitoring and log processing were also part of this environment and were defined as data acquisition and data warehouse modules. The data originated from heterogeneous data sources, e.g., air interfaces, 5G core network control planes, and network management or firewall logs. The data were collected by using the Safe Data Transfer Protocol (SDTP) and Secure File Transfer Protocol (SFTP). However, the authors did not provide any details on data preprocessing tools.

Apache NiFi is a mature tool that has formed the basis of many projects. These range from a project for building a stable data loading system [10] to more demanding ones, such as spatial data analysis [11] or pedestrian safety monitoring [12]. In addition, Apache NiFi is often used  in the context of the Internet of Things (IoT) [13,14]. Interesting applications have included sentiment analysis [15], the facilitation of industrial production optimization [16] (one of the components of the MONSOON project), and personalized smart home automation [17]. Alternatives to Apache NiFi, such as Azure Synapse and Azure Data Factory, were described in [18]. As mentioned in the introduction, NiFi is also a great candidate for gathering information on the state of 5G/6G networks comprising the optical layer. These data are crucial for ensuring the high reliability and availability of the network.

However, it is hardly possible to find research papers that address the use of Apache NiFi in the context of 5G network monitoring. This is considered in some of the deliverables and theses mentioned below. That is, the usage of Apache NiFi for the purpose of collecting measurement and monitoring data was considered in the deliverable [19] of the Horizon 2020 5Growth Project. The project itself was focused on the validation of end-to-end 5G solutions from the perspective of verticals such as Industry 4.0, transportation, and energy. Apache NiFi was used as part of a semantic data aggregator to transport and preprocess the data. The tool was integrated with various data sources, including telemetry-based sources, with a special focus on using YANG models. Apache NiFi was also used to collect static transmission timing interval (TTI) traces from a 5G network for the purpose of troubleshooting and further processing in the proposed architecture [20]. The precise goal of Apache NiFi is to fetch the input TTI traces (in the form of an archive file), retrieve proper files from the archive, and upload them to the cloud. Finally, the author of the master's thesis in [21] proposed a generic architecture that aggregated monitoring data from heterogeneous sources in a 5G infrastructure. The author implemented the architecture by using two different approaches, i.e., NGSI-LD Context Broker and Apache Kafka. In both cases, Apache NiFi was used as a data producer (collecting data from the original source and preprocessing them) and consumer (with receiving notifications being the final output of the system).

One must note that all of these works focused on 5G infrastructure, but were not devoted to the usage of optical networks. Additionally, the mentioned approaches that did not utilize Apache NiFi had to use separate tools to collect and preprocess the data. Finally, the usage of Apache NiFi in the context of optical infrastructure for a 5G network

has been considered, but has not been carefully investigated or validated in any research papers. Thus, to the best of our knowledge, this is the first research paper to simultaneously propose, optimize, and experimentally validate an application of Apache NiFi for the purpose of monitoring an optical network in a 5G infrastructure.

## 4. Commercial Deployment of NiFi to Monitor a 5G Optical Network

In this section, we present how Apache NiFi supports the monitoring of optical infrastructure in a 5G network. We focus on a particular deployment as an example. However, an analogous solution can be applied to 6G infrastructure. Figure 1 illustrates the production environment. Statistics from optical transceivers are collected and aggregated to different Kafka topics by the client's internal solution. These steps could be easily incorporated into the NiFi. The Apache NiFi block represents the entire data processing flow. In this work, we focus on the performance of this part. A reference database (DB) stores the results. Each row consists of such information as an identifier, model, serial number, or last update date. These data are later used to deliver services requested by the client—for example, a graphical representation of the optical infrastructure in the form of a tree hierarchy. As a whole, the delivered solution applies to multiple 3GPP and tmforum specifications—for example, those for management and orchestration [22,23]. The data model is based on [24].

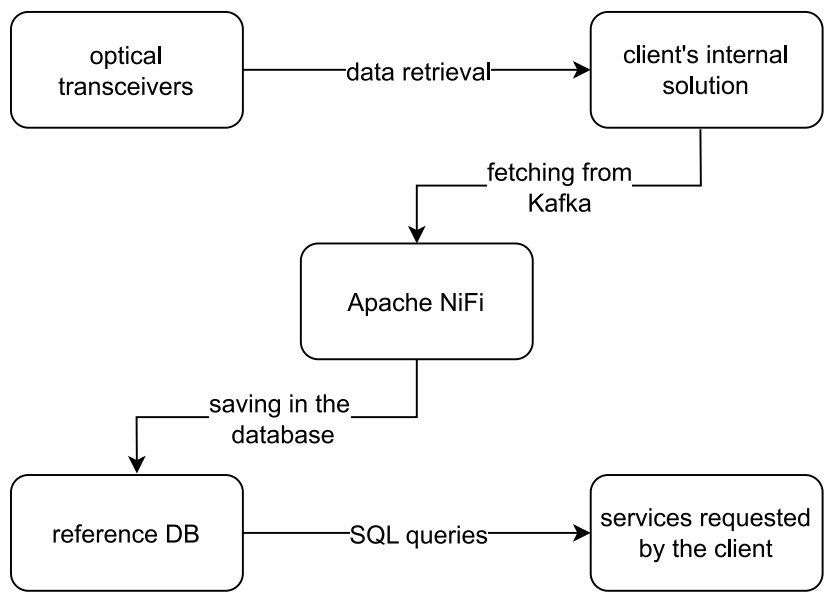

**Figure 1.** Architecture of the commercial solution.

The first task performed by NiFi was that of retrieving data from various Kafka (https://kafka.apache.org/, accessed on 13 February 2023) topics that were aggregating information about different elements in the optical network, i.e., devices and transceivers. A dedicated topic was assumed for each network interface card. Due to the high configurability, NiFi was able to process all of the data without the need for additional dedicated services. Instead, the information from heterogeneous optical devices was dynamically routed to different process groups, which handled object types that were specific to particular devices and vendors. As a result, the platform can be easily modified and extended. It is also resilient to processing errors, as flows are independent of each other. NiFi is also capable of performing the active approach that we investigated in our work. Namely, the SNMP can be used to automatically receive logging data from optical devices in a 5G network (represented as datasets in our work). Part of our solution can be viewed in Figure 2, which presents the system components responsible for loading the data.

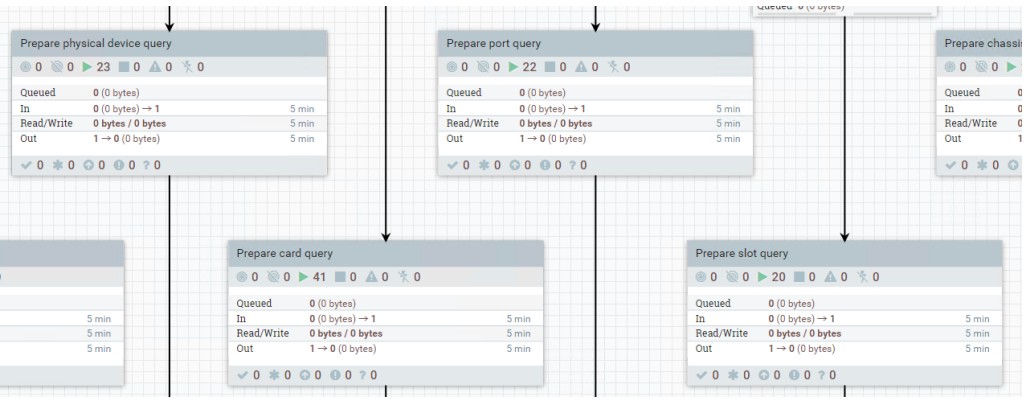

**Figure 2.** FlowFile Processors responsible for loading the data into Apache NiFi.

In the next step, NiFi extracted some fundamental information about fetched objects—for example, the identifier and the type of requested operation (e.g., create, update, or delete). Then, FlowFiles were processed by the modules to provide a mapping between the JSON format (data loaded from Kafka) and an SQL statement. Figure 3 presents the mapping process. Each processor group (grouped FlowFile Processors; here, they are labeled *Prepare ... query*) was dedicated to a different object type. Based on the value of the *type* field, the *RouteOnAttribute* FlowFile Processor decided about passing a FlowFile to the proper mapping block.

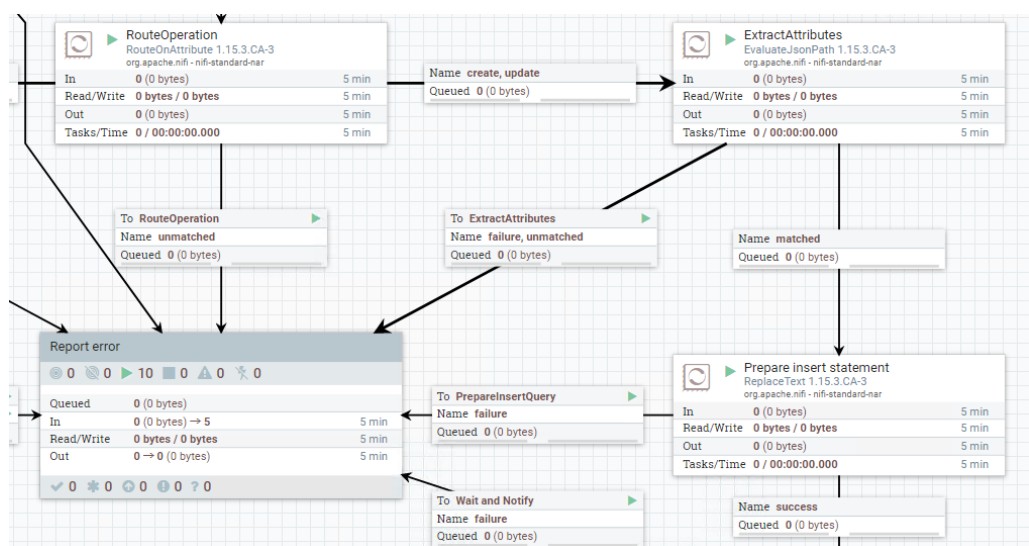

**Figure 3.** Blocks responsible for mapping.

Each process group consisted of three main parts. The first one prepared DELETE statements, the second logged processing errors, and the final part (presented in Figure 4) handled the CREATE/UPDATE requests (these types came from the architecture of the whole commercial system; their names refer to an operation that must be performed on an object specified in a message).

**Figure 4.** Fragment of a mapping block dedicated to the object type "Port" (optical transceiver).

Figure 4 presents the universal logic because each object type had a similar CREATE/UPDATE part that consisted of two main components. First, the *ExtractAt-*

*tributes* Processor was responsible for extracting values of the defined JSON fields and saving them as the attributes of a FlowFile. The attribute name would be taken from the *Property* column. The value would be taken from the JSON field specified by the JSON path in the *Value* column. Later, these newly created attributes would be utilized by the *Prepare insert statement* FlowFile Processor to compose an appropriate SQL statement. An example of the FlowFile attributes can be found in Figure 5.

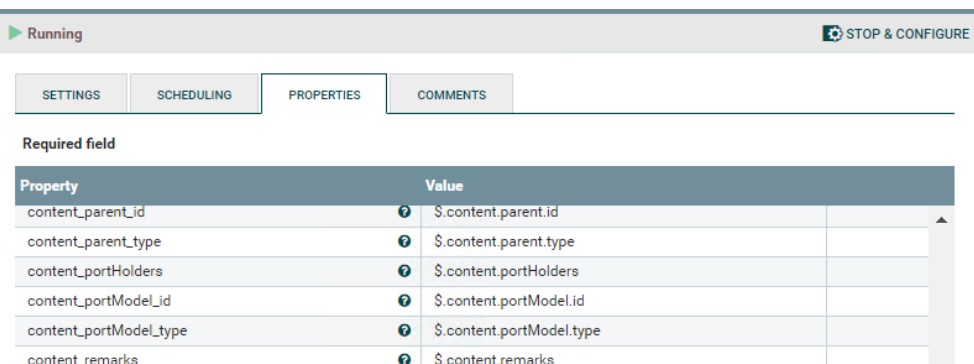

**Figure 5.** Attributes' creation.

Once the required data were extracted, the contents of the original JSON were no longer needed and could be overwritten with the SQL statement, which was necessary in order to upload new data into the reference database. This task was performed by *Prepare insert statement*. Part of its configuration is shown in Figure 6. It stipulates that NiFi will overwrite the original JSON with the replacement value ${*attribute_name*} in the SQL template (the ${} notation is a part of NiFi Expression Language (https://nifi.apache.org/docs/nifi-docs/html/expression-language-guide.html, accessed on 13 February 2023), which is a simple scripting language; here, it is used to fetch the value stored in a particular attribute).

```
 4                   WHEN MATCHED THEN UPDATE
 5                     SET
 6  p.con_additionalAttributes = '${content_additionalAttributes}'
 7  , p.con_auditInfo_cre_id = '${content_auditInformation_created_id}'
 8  , p.con_auditInfo_cre_links = '${content_auditInformation_created_links}'
 9  , p.con_auditInfo_cre_time = '${content_auditInformation_created_time}'
10  , p.con_auditInfo_cre_traceId = '${content_auditInformation_created_traceId}'
11  , p.con_auditInfo_cre_userId = '${content_auditInformation_created_userId}'
12  , p.con_auditInfo_upd_id = '${content_auditInformation_lastUpdated_id}'
13  , p.con_auditInfo_upd_links = '${content_auditInformation_lastUpdated_links}'
14  , p.con_auditInfo_upd_time = '${content_auditInformation_lastUpdated_time}'
15  , p.con_auditInfo_upd_traceId = '${content_auditInformation_lastUpdated_traceId}'
16  , p.con_auditInfo_upd_userId = '${content_auditInformation_lastUpdated_userId}'
17  , p.content_description = '${content_description}'
18  , p.content_device_type = '${content_device_type}'
```

**Figure 6.** Preparation of the SQL statement.

Afterward, the previously prepared SQL statements were merged in batches of 1000 to avoid overloading the database with too many transactions and to improve performance. A deployment scale can be observed in Table 1 (this system is still in the development stage as of 05.10.2022; thus, not every physical object has been uploaded to the system, and the final object count will be larger). With so many optical network components, the monitoring system has to be greatly optimized to stay responsive, robust, and consistent. It is especially important for demanding 5G applications and will be even more critical in the context of 6G. This might become a challenge to a great extent when there are heterogeneous optical devices in a physical network.

**Table 1.** Summary of the data processing.

|  | **Physical Device** | **Optical Port** | **Trail** |
|---|---|---|---|
| Kafka (input) | 95,894 | 1,536,536 | 291,796 |
| Reference DB (output) | 1935 | 49,524 | 13,945 |

The data concerning optical transceivers were stored in a single Kafka topic in the form of 1,536,536 JSON files. It took around 4 min to load the data, process them in Apache NiFi, and save the results in the reference database. The execution time was estimated due to the constantly increasing size of the data (each change of state of a given network element generated a new Kafka message). The typical time delay between publishing a new JSON file to the Kafka topic and saving it in the Apache NiFi database was less than 1 min. This is fully sufficient for operators to reflect changes in the optical topology when serving 5G deployments and will also be acceptable in the case of 6G.

## 5. Research and Results

As the environment presented in Section 4 serves commercial and production purposes, we were not able to utilize it to conduct our research. Furthermore, we also had to use other datasets to keep the original data confidential. However, despite these differences, our work can be used directly in the context of monitoring optical networks in 5G/6G deployments.

All experiments were carried out on the CORD-19 dataset (https://www.kaggle.com/datasets/allen-institute-for-ai/CORD-19-research-challenge?select=document_parses, accessed on 13 February 2023). We investigated the effect of the number of nodes in a cluster on the performance of Apache NiFi. The task carried out by the Apache NiFi instance under study was that of extracting a unique list of scientific papers cited in the articles included in CORD-19. Among other things, the title, authors, and publication date were extracted. The input data were in JSON format, and the results were saved as csv files. This experimental dataset can be easily used to benchmark analogous operations in the context of optical network monitoring in 5G production environments. The only difference will be changing the titles and dates of the research papers into the selected parameters extracted from the log files collected from optical devices. Thus, the particular content of each file is not critical. We chose this particular dataset to perform our experiments because it is easily available at no charge and is well documented.

Regardless of the data context, the processing path was divided into three parts, as shown in Figure 7. The first loaded the data, the second performed the actual processing, and the last saved the results. Detailed documentation and configuration files, the configuration of each individual FlowFile Processor, and the connections between them were placed in a git repository (https://github.com/karolwn/Apache_NiFi, accessed on 13 February 2023).

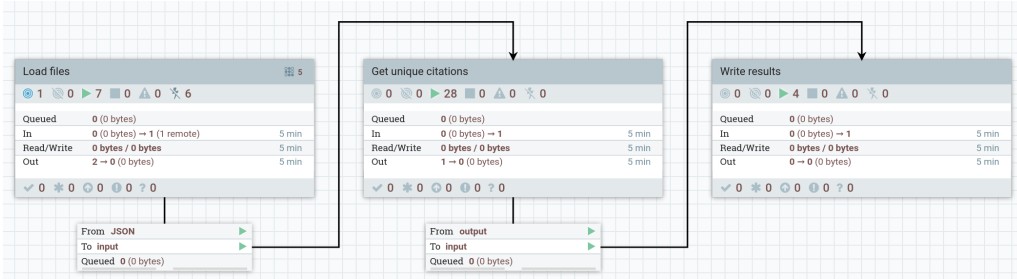

**Figure 7.** Configured processing path.

Figure 8 presents the logical scheme of the experiment, which was as follows. The downloaded CORD-19 dataset was trimmed to 200,000 JSON files (17.7 GB). This was done to reduce the required disk storage space. This subset was later compressed, and before

each iteration of the experiment, it was extracted to the shared input folder. Apache NiFi read data from this folder and processed them, as shown in Figure 7. The obtained results were saved in a different shared folder to prevent input and output data from mixing. In the final step, performance statistics were calculated based on the processed entries and additional information about the utilization of virtual machines (the script is available in the git repository (https://github.com/karolwn/Apache_NiFi, accessed on 13 February 2023)).

In this work, we propose an original modification of the Apache NiFi configuration to improve the data processing speed. To achieve faster processing, we first suggest modifying the default values of the Concurrent Tasks and Run Duration parameters that are equal to 1 and 0 ms, respectively. Table 2 presents the proposed values for consecutive processors according to the order in the processing path. Furthermore, within the module group responsible for data loading, the maximum queue size was reduced from 1 GB to 200 MB to save RAM. On the other hand, within the group responsible for data processing, the limit of cached FlowFiles was increased from the default of 10,000 to 30,000. The proposed improvements are an original contribution of this publication. Individual settings were chosen experimentally and iteratively to achieve error-free system operation with the highest possible performance. A similar optimization process was performed in the aforementioned commercial 5G environment on the components of the system that handled the relations between objects. This was required due to the following properties of the source data in the production environment. Typically, one server had four optical transceivers and six twisted-pair ports that generated events for the system. Furthermore, a single virtualization hypervisor could run multiple virtual machines, and each of the virtual machines could host multiple containers. As a result, one FlowFile could even generate up to 20 new FlowFiles. This became an issue during full resynchronization of the data because the entire Kafka topic was loaded in a matter of seconds, which would flood the processing path. As a countermeasure, we increased the size of queues from the default of 10,000 to 100,000 or 200,000 depending on the particular case.

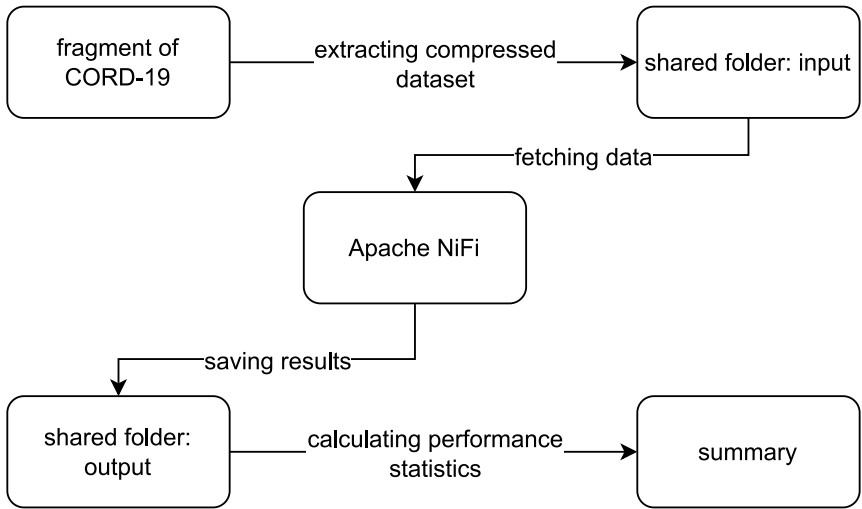

**Figure 8.** The logical scheme of the experiment.

**Table 2.** Proposed parameter values for individual FlowFile Processors.

| Processor | Concurrent Tasks | Run Duration [ms] |
| --- | --- | --- |
| GetFile | 10 | N/A |
| NiFiFlow | 10 | N/A |
| EvaluateJsonPath | 10 | 500 |
| UpdateAttribute | 10 | 500 |
| JoltTransformJSON | 7 | 1000 |
| EvaluateJsonPath | 2 | 1000 |
| SplitJson | 3 | 500 |
| EvaluateJsonPath | 10 | 2000 |
| ExtractAuthors | 5 | 2000 |
| UpdateAttribute | 5 | 2000 |
| ReplaceText | 10 | 2000 |
| HashContent | 5 | 2000 |
| DetectDuplicate | 12 | 2000 |
| ReplaceText | 10 | 2000 |
| MergeContent | 5 | N/A |
| ReplaceText | 5 | 50 |
| UpdateAttribute | 1 | 0 |
| PutFile | 5 | 25 |

All tests were run on virtual machines hosted by the type 2 hypervisor in Oracle VirtualBox version 6.1.26. Each was assigned two AMD Ryzen 9 5900X processor cores at an average clock speed of 4.4 GHz, 4256 MB of RAM at 3600 MHz, and 60 GB of storage space (NVMe PCIe 3.0 drive). The test and result sets were stored in a folder shared among all hosts (NVMe PCIe 4.0 drive). We used Ubuntu Linux 20.04 LTS as an operating system. We monitored the following parameters: the load average (average load on the Linux system), CPU utilization percentage, RAM occupancy, and processing time for both a single file and the entire dataset. In consecutive scenarios, the clusters consisted of a number of nodes varying from 1 to 5 (with a step of 1). The rationale was that scalability is a critical issue for making Apache NiFi usable in optical networks that support 5G/6G networks due to the possible deployments of new sites that may be composed of numerous optical devices reporting operational parameters. Adding nodes can be the only way to handle the additional load. Detailed results are presented for scenarios with one, three, and five nodes in a cluster to analyze the performance as a function of cluster size. We repeated every experiment eight times to ensure statistical credibility. In general, the results are presented in graphs, and the numerical values in Table 3 are presented with 98% confidence intervals. However, due to the high variability of the system load parameters, the confidence intervals for experiments with more than one node in a cluster were extended to 68%.

*5.1. One Node*

Figure 9 shows the variation in the load average parameter value during the data processing. The different line styles denote results averaged from the last one, five, and fifteen minutes. The vertical dashed unlabeled line in all graphs indicates when the optimized node finished processing the input set against the default configuration. The load average values of the optimized node oscillated around a value of 8.5, while for the default case, the corresponding value was equal to 7. This difference resulted from the higher average CPU usage of the node for the configuration proposed in this article. The fluctuations in the one-minute load average were caused by the scheduler settings of the individual FlowFile Processors because they did not run continuously. Instead, they were triggered by the scheduler. Wide confidence intervals in the case of the default configuration demonstrated the unstable operation of the infrastructure. For the default configuration, the values of the load average for even a 15-min period ranged from 4 to 9 between the 60th and 160th minutes of the experiment. This discrepancy resulted from the completely full queues, which caused interruptions in the runtime of the FlowFile

Processors. The proposed optimization eliminated these problems and ensured the correct and stable operation of the system. Thus, the proposed improvements enabled the use of Apache NiFi in 5Gproduction networks by ensuring its reliability under demanding conditions. In the future, an analogous solution will also be applicable to 6G infrastructure.

The individual residence times of the FlowFile in the system, i.e., the times needed to extract a single resulting entry, are shown in Figure 10. The proposed optimization reduced the average processing times by approximately one order of magnitude from 1406 to 102 s. This means that the state of the 5G infrastructure reported by the optical network devices could be processed up to ten times faster than in the default configuration. It further significantly improved the effectiveness of monitoring of the complex environment. For example, information about failures in the optical layer of the 5G/6G network will be delivered to the NOC (network operations center) 10 times faster. This could significantly reduce the impact of any events and, thus, the potential costs for the operators of 5G and future 6G networks. Additionally, the response time for any failure could be shortened, which could positively impact the overall quality of service (QoS) offered to the client.

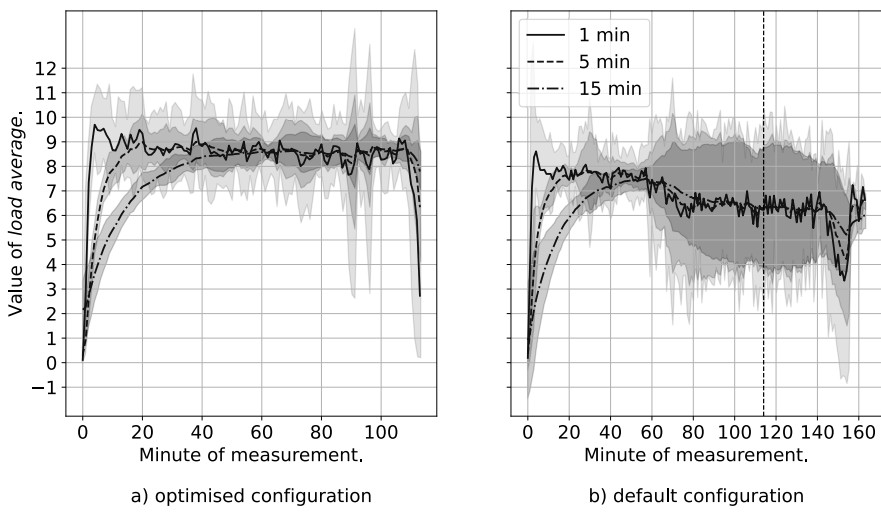

**Figure 9.** Load average parameter as a function of the processing time of the input set (one node). Panel (**a**) shows the result of the optimized configuration, whereas (**b**) depicts the default configuration.

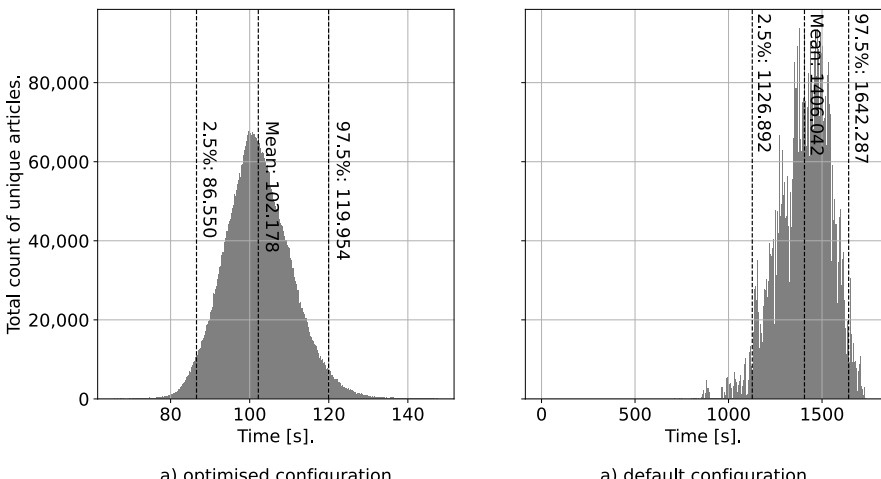

**Figure 10.** Distribution of time required to extract a single cited article (one node). Panel (**a**) shows the result of the optimized configuration, whereas (**b**) depicts the default configuration.

Table 3 presents the utilization of CPU and RAM as a function of the cluster size for both the default and optimized configurations. The Apache NiFi instance that was

subjected to optimization used the server's CPU resources approximately 30 percentage points more efficiently compared to the default configuration when using a single node. Simultaneously, the memory usage was kept at a similar level. This should be explained by the fact that the Java Virtual Machines (JVMs) had the same amount of RAM available. The increased CPU consumption in the case of the default configuration was related to the numerous I/O operations triggered by overloaded queues. More precisely, Apache NiFi saved operating memory by writing the FlowFiles to the disk. Thus, numerous high-density optical devices running in 5G/6G networks can easily over-utilize an Apache NiFi deployment in the default configuration. A regular-sized deployment may consist of a few hundred thousand components to be monitored. Due to the proposed improvements, it was possible to eliminate the queue overflow issue and enable the use of Apache NiFi in complex optical networks that support 5G infrastructures. In the future, this can be also examined with respect to the requirements of 6G networks.

**Table 3.** Numerical results.

| Node Number | CPU—User [%] | CPU—System [%] | RAM Used [MB]. |
|:---:|:---:|:---:|:---:|
| One Node—Optimized Configuration | | | |
| 1 | 81.36 ± 11.65 | 13.01 ± 1.54 | 3005.48 ± 125.98 |
| One Node—Default Configuration | | | |
| 1 | 49.25 ± 6.37 | 30.1 ± 6.14 | 3006.04 ± 105.79 |
| Two Nodes—Optimized Configuration | | | |
| 1 | 74.13 ± 14.62 | 14.64 ± 2.79 | 3015.77 ± 118.18 |
| 2 | 73.86 ± 15.86 | 13.91 ± 2.89 | 2813.26 ± 155.01 |
| average | 73.99 ± 15.24 | 14.27 ± 2.84 | 2914.51 ± 136.6 |
| Two Nodes—Default Configuration | | | |
| 1 | 40.45 ± 11.16 | 22.01 ± 6.85 | 2910.07 ± 134.71 |
| 2 | 37.06 ± 13.32 | 21.47 ± 8.26 | 2624.18 ± 179.29 |
| average | 38.76 ± 12.24 | 21.74 ± 7.55 | 2767.13 ± 157.0 |
| Three Nodes—Optimized Configuration | | | |
| 1 | 65.3 ± 18.17 | 14.6 ± 4.11 | 2966.05 ± 145.41 |
| 2 | 67.09 ± 19.59 | 15.2 ± 3.71 | 2773.77 ± 195.91 |
| 3 | 69.23 ± 17.27 | 14.05 ± 3.26 | 2781.91 ± 198.71 |
| average | 67.21 ± 18.34 | 14.62 ± 3.7 | 2840.58 ± 180.01 |
| Three Nodes—Default Configuration | | | |
| 1 | 52.23 ± 11.89 | 29.92 ± 9.58 | 2846.2 ± 157.52 |
| 2 | 46.07 ± 12.96 | 26.53 ± 9.63 | 2532.66 ± 254.31 |
| 3 | 48.87 ± 11.35 | 27.19 ± 8.75 | 2588.47 ± 213.04 |
| average | 49.06 ± 12.07 | 27.88 ± 9.32 | 2655.78 ± 208.29 |
| Four Nodes—Optimized Configuration | | | |
| 1 | 62.1 ± 18.63 | 15.23 ± 4.36 | 2919.01 ± 203.6 |
| 2 | 63.47 ± 18.62 | 14.29 ± 3.89 | 2733.87 ± 240.76 |
| 3 | 62.87 ± 18.25 | 13.54 ± 3.86 | 2728.67 ± 248.5 |
| 4 | 63.76 ± 17.7 | 14.26 ± 4.6 | 2748.53 ± 243.51 |
| average | 63.05 ± 18.3 | 14.33 ± 4.18 | 2782.52 ± 234.1 |

**Table 3.** *Cont.*

| Node Number | CPU—User [%] | CPU—System [%] | RAM Used [MB]. |
|---|---|---|---|
| | Four Nodes—Default Configuration | | |
| 1 | 51.07 ± 10.71 | 27.35 ± 10.69 | 2865.79 ± 184.68 |
| 2 | 50.44 ± 11.5 | 26.87 ± 11.08 | 2672.01 ± 231.48 |
| 3 | 48.89 ± 14.16 | 26.89 ± 11.45 | 2644.76 ± 240.33 |
| 4 | 36.89 ± 15.86 | 22.91 ± 11.2 | 2528.87 ± 326.95 |
| average | 46.82 ± 13.06 | 26.01 ± 11.1 | 2677.86 ± 245.86 |
| | Five Nodes—Optimized Configuration | | |
| 1 | 58.18 ± 18.88 | 13.36 ± 4.05 | 2939.28 ± 225.73 |
| 2 | 57.75 ± 19.17 | 12.55 ± 3.35 | 2725.8 ± 247.69 |
| 3 | 60.96 ± 17.13 | 12.85 ± 3.85 | 2734.32 ± 268.28 |
| 4 | 59.38 ± 19.96 | 13.23 ± 4.38 | 2729.3 ± 261.57 |
| 5 | 59.0 ± 18.81 | 13.79 ± 4.29 | 2736.59 ± 264.59 |
| average | 59.05 ± 18.79 | 13.15 ± 3.99 | 2773.06 ± 253.57 |
| | Five Nodes—Default Configuration | | |
| 1 | 41.58 ± 21.67 | 22.83 ± 13.6 | 2990.34 ± 183.84 |
| 2 | 51.2 ± 10.69 | 22.58 ± 13.75 | 2666.11 ± 226.08 |
| 3 | 47.91 ± 11.57 | 21.97 ± 13.17 | 2628.47 ± 233.14 |
| 4 | 38.34 ± 21.47 | 22.24 ± 13.48 | 2628.78 ± 234.11 |
| 5 | 49.63 ± 10.74 | 22.96 ± 12.59 | 2641.24 ± 227.66 |
| average | 45.73 ± 15.23 | 22.52 ± 13.32 | 2710.99 ± 220.97 |

*5.2. Three Nodes*

A cluster composed of three nodes was able to process the data three times more than a single-node cluster. This performance improvement proved the ability of Apache NiFi to horizontally scale. In both the optimized and default configurations, the mean value of the load average was slightly decreased compared to that in the single-node scenario.

Additionally, the total CPU usage decreased by about 15 percentage points for the optimized configuration, whereas for the default configuration, it stayed on the same level (see Table 3). It is interesting that the confidence interval calculated for the load average parameter in the default configuration significantly decreased (see Figure 11). This suggests that the cluster operated under more stable conditions.

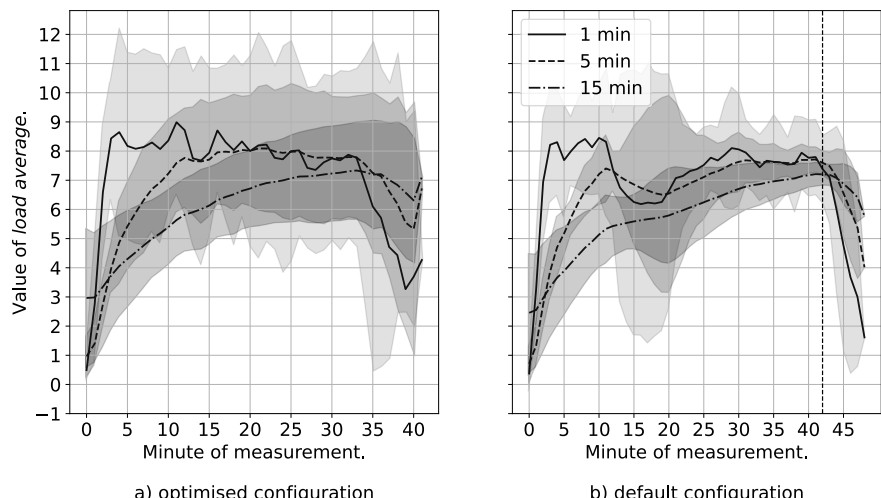

a) optimised configuration                    b) default configuration

**Figure 11.** Load average parameter as a function of the processing time of the input set (three nodes). Panel (**a**) shows the result of the optimized configuration, whereas (**b**) depicts the default configuration.

The average time required to obtain one result entry in the optimized configuration was reduced by about two times (from 102 to 57 s) compared to that in the standalone scenario. In contrast, the analogous parameter in the case of the default configuration was reduced by approximately 400 s from 1400. Detailed histograms are shown in Figure 12. Thus, the optimized configuration consequently achieved significantly better results (57 versus 1000 s on average).

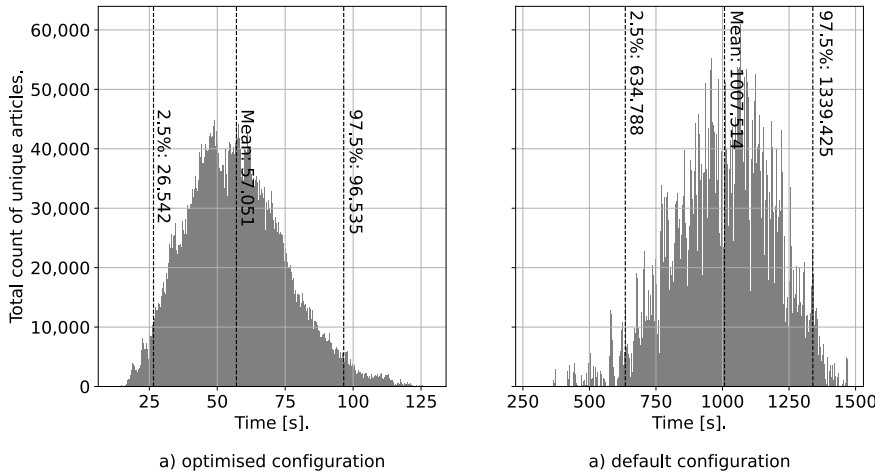

a) optimised configuration      a) default configuration

**Figure 12.** Distribution of time required to extract a single cited article (three nodes). Panel (**a**) shows the result of the optimized configuration, whereas (**b**) depicts the default configuration.

In the NiFi deployment for the commercial 5G network, the cluster also consisted of three nodes. We were unable to extract the raw data related to the load average due to the security policy. As a substitute, in Figure 13, we present the graphical results generated by NiFi. The average value of the load average across the production cluster was lower than that in our research environment. This was mostly due to hardware differences. The machines used during the research had only 4 GB of RAM and two CPU cores, while the operator's machines had 30 GB of RAM that were dedicated exclusively to NiFi and 20C/40T Intel Xeon Gold 6230N CPUs with an average clock rate of 2.3 GHz.

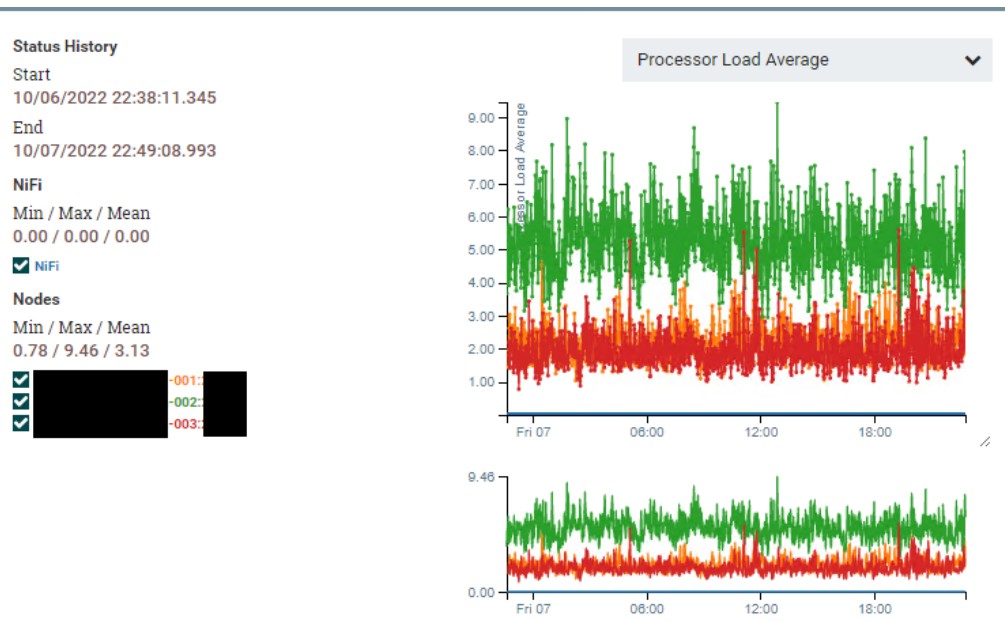

**Figure 13.** Load average extracted from the client configuration.

### 5.3. Five Nodes

Running a cluster of five nodes resulted in a drop in the average CPU usage by user processes of around 10 and 4 percentage points compared with the three-node scenario for the optimized and default configurations, respectively. It should be noted that the variability in resource usage was so high that the statistical differences in the user CPU usage between the configurations were blurred. Detailed results are also presented in Table 3. The memory usage for both configurations decreased by approximately 300 MB relative to that in the single-node scenario.

The cluster consisting of five optimized processing nodes achieved values of the load average parameter that were smaller by approximately one compared to that obtained with the default settings (Figure 14). However, the confidence intervals broadened. Most likely, this was caused by the fact that the FlowFile Processors were not fully utilized during the processing. We believe that this relatively small amount of available RAM started to limit the influx of FlowFiles, which led to idling, and this was further observed as a broad confidence interval.

Running a cluster of five processing nodes allowed a significant reduction in the value of the average time required to obtain one resulting entry (Figure 15). The reductions were approximately 1 and 14 min compared to the single-node instance for the optimized and default configurations, respectively. Nevertheless, the proposed configuration still had a significant advantage. On average, the fine-tuned five-node cluster needed about 40 s, and the default five-node cluster needed up to 534 s to extract a single result entry.

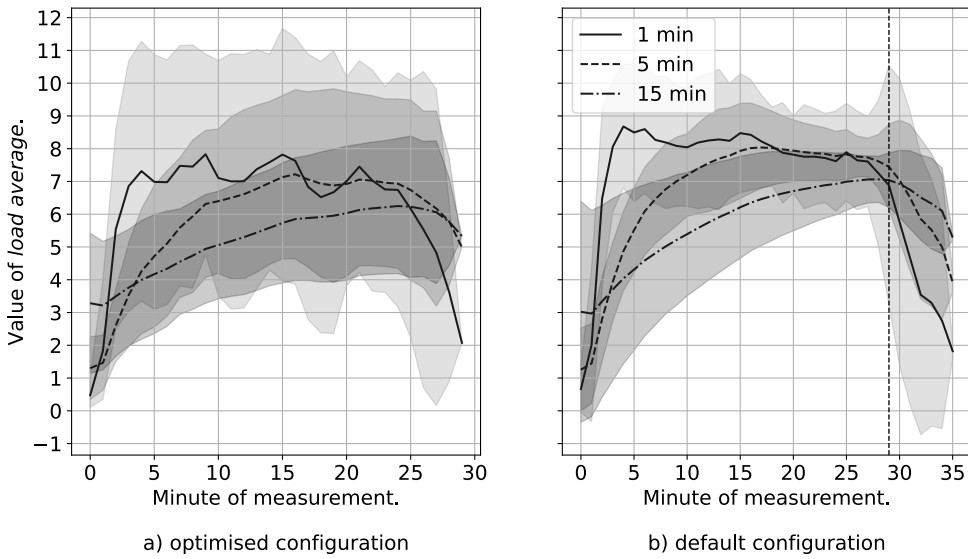

**Figure 14.** Load average parameter as a function of the processing time of the input set (five nodes). Panel (**a**) shows the result of the optimized configuration, whereas (**b**) depicts the default configuration.

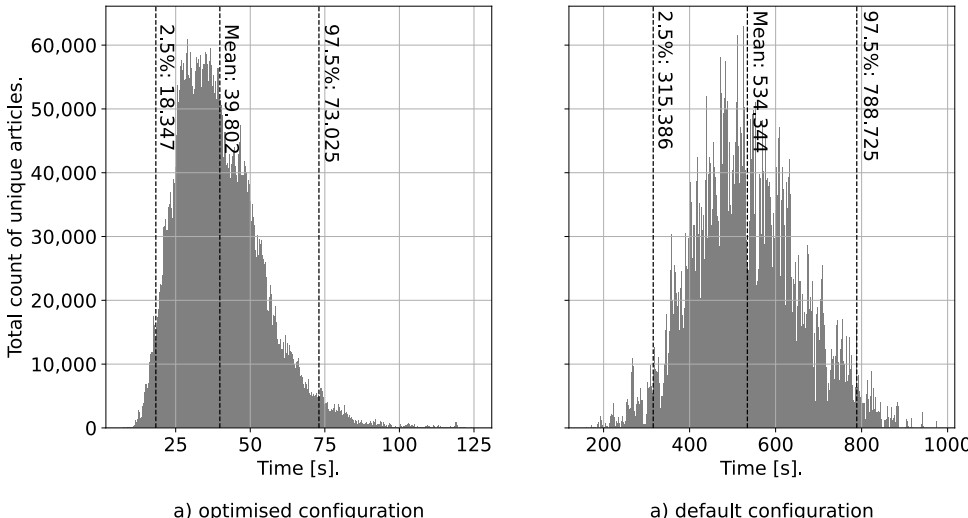

**Figure 15.** Distribution of time required to extract a single cited article (five nodes). Panel (**a**) shows the result of the optimized configuration, whereas (**b**) depicts the default configuration.

To sum up, our optimization reduced the potential operational expenses of a monitoring system for a 5G (and future 6G) network operator. Instead of adding more nodes, it is possible to utilize the existing infrastructure more effectively and to successfully collect monitoring results for the underlying optical network. Furthermore, the improvement in the performance as a function of the number of nodes is less significant for larger clusters if Apache NiFi is not properly configured. In 5G networks and beyond, this may be a limitation, as adding sites will entail adding a significant number of additional optical devices with the large number of high-speed optical transceivers reporting their parameters.

### 5.4. Summary of the Results

Figure 16 shows the total processing time of the input set as a function of the number of nodes in the cluster. In both the default and optimized configurations, the biggest gain was achieved by adding a second node to the cluster. That is, the processing was two times faster. Adding more instances did not bring such a large gain. This effect was caused by the increased overhead required for cluster management and load balancing. Interestingly, the proposed configuration using $n$ nodes allowed the input set to be processed in an amount of time that was close to that required by a cluster of $n + 1$ nodes in the default configuration. It is also important to note the reduction in the difference between the optimized and default configurations after the third node was included in the cluster. This suggests that optimization is especially important for highly utilized clusters.

Efficient cluster scaling is crucial in the case of 5G/6G networks. The increasing number of high-density optical devices incorporated with every additional site will require more computing power to ensure effective monitoring. On the other hand, it is also important not to oversize the cluster. Above some threshold, larger clusters will not bring any significant advantages to the processing speed. For example, based on Figure 16, we can use the Elbow method to conclude that the optimal number of nodes is 3 because there is a minimal gain when enabling the fourth and fifth nodes.

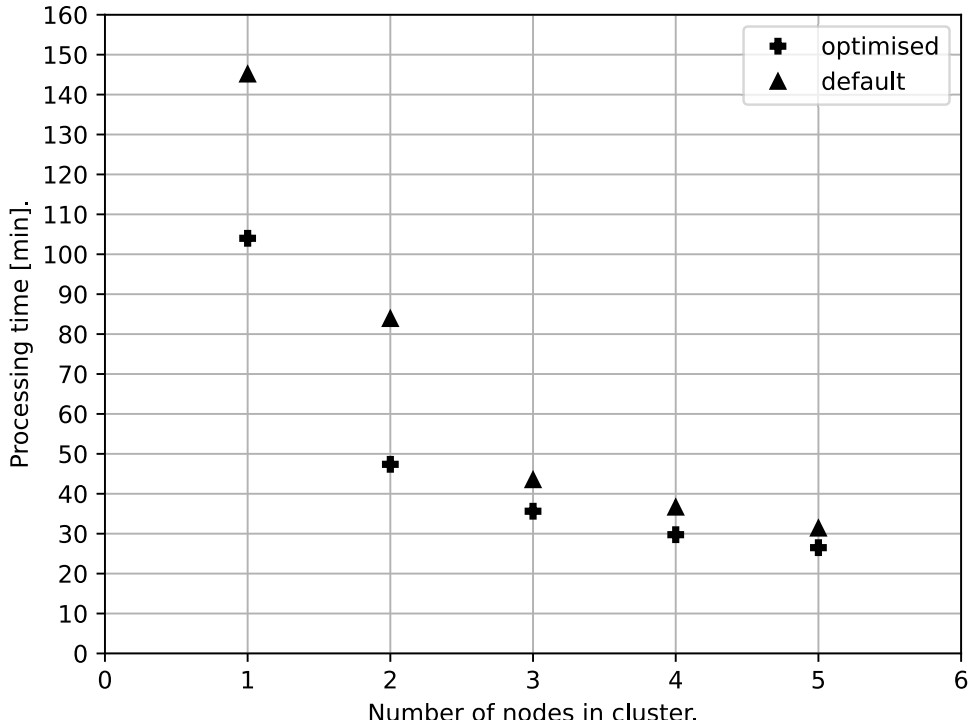

**Figure 16.** Summary of the processing time for the entire collection.

Despite the relatively small differences in the total processing times of the input set, the difference between the default and optimized configurations was even more significant for the time required to serve a single FlowFile. As shown in Figure 17, it reached an order of magnitude. This is the result of the long time for which a FlowFile waits in queues when the system is overloaded in the default configuration.

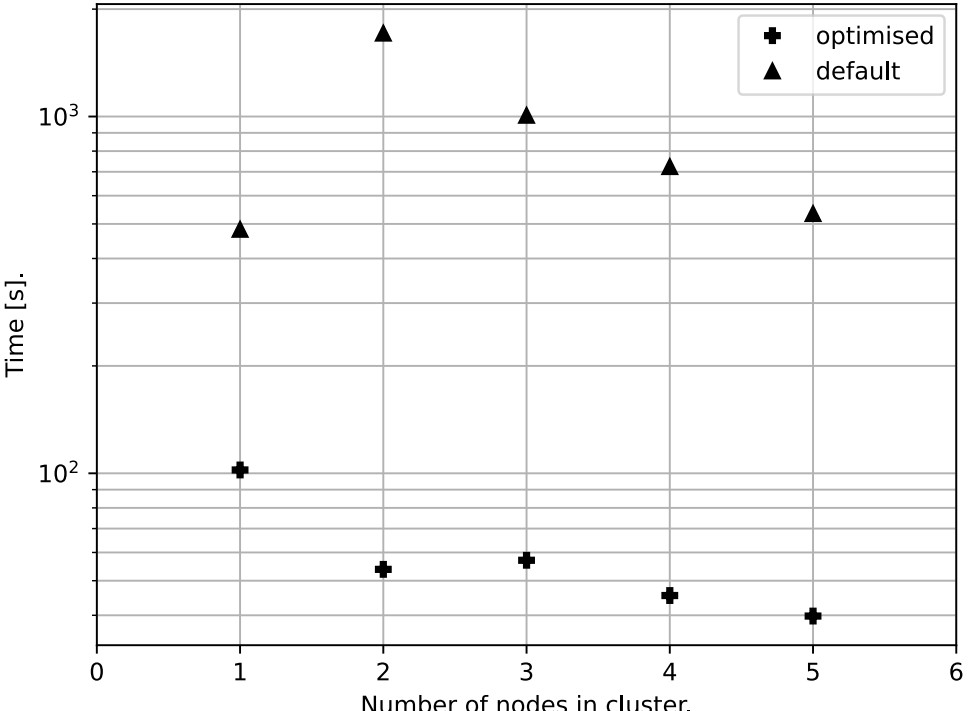

**Figure 17.** Average processing times for individual files.

## 6. Summary

The results presented in this paper show a significant improvement in the performance of the data processing path as a result of the proposed parameter modifications. Due to the authors' configuration, it was possible to achieve shorter processing times for a single unit of data and for the entire collection, as well as much more efficient use of available resources. We benchmarked our proposal on a dataset that is analogous to the data originating from the monitoring of optical networks in 5G architectures and beyond. In our analysis, we focused on the parameters and aspects that are especially important for the operators of optical networks composed of large numbers of enterprise-class high-density optical devices. Thus, we proposed a solution that allows one to effectively gather monitoring data from optical equipment produced by multiple vendors and aggregate them into a single system. Furthermore, our solution significantly reduces both querying times for a single entry from the collected log files and processing times, which are critical for effectively identifying issues occurring in optical networks.

Additionally, it was shown that Apache NiFi's data processing time decreased nonlinearly as the number of nodes in the cluster increased. This change further depended on the parameter configuration. We showed a method based on the analysis of processor utilization to assess if the system was overloaded (mostly for the default configuration) or was still offering the potential for further performance improvements (proposed configuration).

Potential avenues for further work include investigating the contributions of individual modules to resource utilization. Furthermore, we also plan to analyze the dependency on the hardware platform and input dataset. In addition, an implementation of the automatic scaling of clusters depending on the load is an interesting research direction. The size of a cluster, thus, may follow dynamically reconfigurable 5G/6G network slices that may drastically change during their lifetimes. NiFi auto-scaling would require the use of additional tools, such as Ansible or Terraform. An additional challenge is that any change to clusters requires a full reboot of these clusters and a swap of the relevant configuration files. This is unacceptable for 5G/6G production deployments, so some additional mechanisms must be developed to seamlessly change the configuration. Finally, we plan to investigate how information aggregated by NiFi may be used to predict failures of optical nodes deployed in 5G/6G networks. Machine-learning-based algorithms may be used to find correlations between parameters that are collected centrally by NiFi and the need for replacing optical hardware to prevent service disruptions.

**Author Contributions:** K.W.: data curation, investigation, methodology, resources, software, visualization, writing—original draft preparation. P.B.: conceptualization, funding acquisition, methodology, project administration, supervision, validation, writing—review and editing. All authors have read and agreed to the published version of the manuscript.

**Funding:** This work was supported by the Polish Ministry of Science and Higher Education with the subvention funds of the Faculty of Computer Science, Electronics, and Telecommunications of AGH University. Some images are the courtesy of Comarch https://www.comarch.com (accessed on 13 February 2023).

**Data Availability Statement:** Commercial data concerning the 5G network is confidential. CORD-19 can be found at https://www.kaggle.com/datasets/allen-institute-for-ai/CORD-19-research-challenge (accessed on 13 February 2023). The NiFi configuration files and source code used to parse the results can be found at https://github.com/karolwn/Apache_NiFi (accessed on 13 February 2023).

**Acknowledgments:** The code and configuration provided in the GitHub repository are licensed under the Apache License 2.0.

**Conflicts of Interest:** The authors declare no conflict of interest. The funders had no role in the design of the study, in the collection, analyses, or interpretation of data, in the writing of the manuscript, or in the decision to publish the results.

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
