# Peer review of "A Data Processing and Distribution System Based on Apache NiFi"

_photonics, doi:10.3390/photonics10020210_

Round 1

Reviewer 1 Report

The subject of this paper is clearly defined; it is devoted to processing huge amounts of data, as, for example, in 5G/6G networks. This work has an experimental character. It has been shown in it that certain mechanisms applied by the authors can increase the speed of the Apache NiFi software system tested by them. The results provided in the figures enclosed confirm the above findings. 

Reviewer 2 Report

The paper reads well and it is very interesting. I suggest to better relate the usage of Apache NIFI with optical transceivers.

Reviewer 3 Report

1. The introduction section is not laid out. In it, you should indicate the problem/s you are trying to solve and your contribution.

2. The picture representing the logical scheme of the experiment is missing.

3. There is no information about it if anyone tried to solve the problem similarly. The Related Work section is missing.

4. In the text, you say you experimented with 5G/6G infrastructure. You are doing it on 5G infrastructure. It would be correct to write that and emphasize that the examination could be applied to 6G infrastructure in the future.

5. When describing the virtual environment you're creating, mention what type of hypervisor you use.

6. You mention a 5G production environment. Please describe it in more detail and possibly add an illustration so that the readers can understand the essence of your research more easily.

7. The list of references is very thin.

Round 2

Reviewer 3 Report

Thanks for adopting the suggestions.